# *Listeria monocytogenes* Cold Shock Proteins: Small Proteins with A Huge Impact

**DOI:** 10.3390/microorganisms9051061

**Published:** 2021-05-14

**Authors:** Francis Muchaamba, Roger Stephan, Taurai Tasara

**Affiliations:** Institute for Food Safety and Hygiene, Vetsuisse Faculty, University of Zurich, CH-8057 Zurich, Switzerland; roger.stephan@uzh.ch (R.S.); taurai.tasara@uzh.ch (T.T.)

**Keywords:** cold shock protein, cold, osmotic, stress tolerance, virulence, biofilm, *Listeria monocytogenes*

## Abstract

*Listeria monocytogenes* has evolved an extensive array of mechanisms for coping with stress and adapting to changing environmental conditions, ensuring its virulence phenotype expression. For this reason, *L. monocytogenes* has been identified as a significant food safety and public health concern. Among these adaptation systems are cold shock proteins (Csps), which facilitate rapid response to stress exposure. *L. monocytogenes* has three highly conserved *csp* genes, namely, *cspA*, *cspB*, and *cspD*. Using a series of *csp* deletion mutants, it has been shown that *L. monocytogenes* Csps are important for biofilm formation, motility, cold, osmotic, desiccation, and oxidative stress tolerance. Moreover, they are involved in overall virulence by impacting the expression of virulence-associated phenotypes, such as hemolysis and cell invasion. It is postulated that during stress exposure, Csps function to counteract harmful effects of stress, thereby preserving cell functions, such as DNA replication, transcription and translation, ensuring survival and growth of the cell. Interestingly, it seems that Csps might suppress tolerance to some stresses as their removal resulted in increased tolerance to stresses, such as desiccation for some strains. Differences in *csp* roles among strains from different genetic backgrounds are apparent for desiccation tolerance and biofilm production. Additionally, hierarchical trends for the different Csps and functional redundancies were observed on their influences on stress tolerance and virulence. Overall current data suggest that Csps have a wider role in bacteria physiology than previously assumed.

## 1. Introduction

The Gram-positive bacterium *Listeria monocytogenes* is a significant public health threat, resulting in several foodborne listeriosis outbreaks that have claimed many lives over the last few decades [1,2,3,4]. It is ranked as one of the most severe zoonoses or foodborne pathogens with one of the highest hospitalizations and mortality rates [3]. In addition, it is a major economic burden for food manufacturers not only through food recalls but also the extensive control measures required to minimize its occurrence and persistence in foods and food processing environments [1,5]. For example, for the United States of America, the economic burden of *L. monocytogenes* is estimated to be over 2.3 billion per year [6]. Due to an aging population and increased consumption of ready-to-eat (RTE) foods, the threat posed by *L. monocytogenes* continues to grow [7,8]. With this in mind, several interventions, such as lowering the water activity, adding protective cultures and other antimicrobials, such as nisin and β-phenylethylamine, have been developed to mitigate the risk of *L. monocytogenes* [9,10,11,12,13]. Several cleaning and decontamination procedures have been employed in the processing environment, however, outbreaks frequencies have largely remained constant [1,5,10,12].

Despite increased knowledge and expertise in good manufacturing practice and hygienic design, *L. monocytogenes* continues to be a recurring food contamination problem [1,2,3,4]. For a microbe to survive in any habitat, it must be able to effectively sense and adapt to adverse ever-changing environmental conditions. Mechanisms that permit stress tolerance, nutrient acquisition and utilization, transcription and translation preservation as well as maintaining protein function must be established. *L. monocytogenes* has evolved multifaceted systems that allow it to switch between the saprophyte and the intercellular pathogen states [14,15,16,17]. Within these two states, a plethora of mechanisms is deployed to facilitate stress tolerance and persistence in the environment, such as disinfectant efflux pumps or biofilm formation [1,16,18,19,20]. More troubling is its ability to grow at low temperatures even under osmotic stress, approaches, which are key hurdle techniques in the food industry. While in the host, the well-timed expression of virulence factors, such as listeriolysin O (LLO) and internalins, ensures evasion of the host immune defenses and effective establishment of infection [17]. In order for this “lifestyle” to be maintained, effective sensing for the distinction between the saprophyte and intercellular states is needed [16,17]. Equally important, judicious regulation of this arsenal of mechanisms is required to ensure that the bacteria thrive in either environment [16,17]. Quorum sensing, two-component systems, and regulators, such as the alternative sigma factor SigB (σ^B^), PrfA, regulatory RNA, and cold shock domain family proteins (Csps), are some of the major sensors and regulators of these adaptation systems [9,10,12,21,22,23,24].

Csps are RNA- and DNA-binding proteins used in bacteria for regulating expression of numerous genes, including those relevant in virulence and stress responses [25,26,27,28,29,30]. Our understanding of how Csps are involved in bacterial stress response and virulence regulation is, however, presently limited. Current paradigms presume that these proteins contribute to the global regulation of transcription and translation processes in bacteria and other organisms [25,30,31,32].

In *L. monocytogenes* deletion mutagenesis of *csp* genes in the reference strain, EGDe has so far been undertaken, indicating that Csps are relevant for stress resistance and virulence [26,27,28,33]. These attributes are central to the food safety and public health issues associated with *L. monocytogenes*. Thus, it is important that our understanding of Csps roles in these phenotypes is improved. An existing limitation is that the strain EGDe, so far employed in Csp studies, represents a *L. monocytogenes* genotype rarely involved in human clinical illnesses [34]. Going forward, the examination of Csp roles in the context of more clinically important and highly virulent *L. monocytogenes* genetic backgrounds is, therefore, needed. Improved understanding of Csp cellular functions in *L. monocytogenes* requires the identification and characterization of the Csp regulated genes, which also has not yet been done in this pathogen. This review focuses on knowledge of Csp-family proteins functions, with emphasis on what is currently known about their role in stress response and virulence in *L. monocytogenes*.

## 2. Csps in *L. monocytogenes*

Csps are low-affinity RNA or DNA-binding proteins, that although initially discovered as key cold-induced proteins responsible for cold stress tolerance, are now recognized to be important in regulating a broad range of gene expression and physiological responses in bacteria [25,28,30,35,36,37,38]. Despite being highly conserved and sometimes having the same name, Csps have functionally evolved to serve diverse roles in bacteria. Three Csps, namely CspA (CspL), CspB and CspD encoded through *lmo1364* (*cspA*), *lmo2016* (*cspB*), and *lmo1879* (*cspD*) genes are found in *L. monocytogenes*.

*L. monocytogenes* Csps are small proteins (66 amino acids in length), containing two RNA-binding motifs (RNBP) similar to those described in other bacteria [39,40,41]. Csps and their RNBP are highly conserved between different genetic backgrounds in this bacterium (Figure 1) [39,40,41]. Overall amino acid sequence identity between the *L. monocytogenes* Csps ranges from 67 to 73% [33]. The limited amino acid differences in the RNBP might contribute to Csp specificity and affinity for particular RNA or DNA regions, which could be responsible for specialization or distinctions in Csp roles. In *Bacillus subtilis*, single amino acid substitutions within the RNBP region resulted in abolished (F15A, F17A, F27A, H29Q) or increased (F15Y) nucleic acid binding capacity of CspB [42]. In *Staphylococcus aureus*, amino acid substitution outside this region in CspA (P58E) resulted in the loss of its biological function to induce staphyloxanthin production, while E58P substitution in CspC resulted in it gaining CspA biological functions [43].

*L. monocytogenes* Csps are characterized by an acidic isoelectric point and a low melting temperature of around 40 °C (CspA) [40]. The tertiary protein structure has been solved for *L. monocytogenes* CspA, showing that their crystal structure comprises of five antiparallel β-strands, which together form a β-barrel structure [40]. This Csp structure is highly conserved among different organisms including bacteria (reviewed by [31,44,45]).

In terms of their molecular functional mechanisms, one of the major functions postulated for Csps is that they act as RNA chaperones minimizing or melting mRNA secondary structures, thereby facilitating initiation and progression of protein translation under adverse conditions, such as low temperatures. Csps are also presumed to function as transcriptional activators. In *Escherichia coli*, CspA has been shown to act as a transcriptional activator for GyrA and H-NS. It is proposed that this effect might be through stabilization of the open complex formation by RNA polymerase as well as acting as transcription antiterminators by preventing the formation of hairpin structures, thereby allowing for full-length mRNA transcription [44,46,47,48]. Some studies also suggest that Csps are involved in mRNA turnover by affecting mRNA stability [27,49,50,51]. Additionally, some have suggested their involvement in DNA replication and chromosomal condensation processes [49,51,52,53,54]. At the protein and ribosomal level, Csps are postulated to assist in refolding damaged proteins and counteract the harmful effects of stress on ribosome functions [41]. Csps have traditionally been known for their role in cold shock responses, hence the name. However, deletion of *csp* genes caused varied phenotypic effects in different bacterial species [41]. For example, deleting all three *csp* genes results in a lethal phenotype in *B. subtilis* but not in *L. monocytogenes* [33,55]. It has also been reported that Csps have functions during normal growth at optimal growth temperature [56].

## 3. Role of Csps in *L. monocytogenes* Stress Tolerance

From its natural habitat and passage through the food processing environment until disease causation in the host, *L. monocytogenes* overcomes several stress hurdles. These range from nutrient starvation, UV light, osmotic stress, bacteriocins, cleaning and disinfectant chemical stress in the environment to pH, lysozyme, bile, and oxidative stress in the host [10,22,57]. *L. monocytogenes* consequently must deploy several stress response systems to adapt and overcome these ever-changing adverse challenges. Here we review the roles of Csps in these processes or against these stressors as well as Csps contributions to virulence (Figure 2).

### 3.1. Cold Stress

Temperature, either high or low, is an important food preservation strategy; low temperatures usually extend product shelf life, while high temperatures are often used to inactivate bacteria. It stands to reason that resistance to such stressors is key for foodborne pathogens. With *L. monocytogenes* being a psychrotroph, its ability to grow at low temperatures is key for it to achieve infectious dose numbers on contaminated refrigerated foods [15,58]. Understanding the mechanisms behind this phenotype is crucial in developing new interventions against this pathogen in refrigerated foods.

Proteome and transcriptome analyses that have been conducted on different *L. monocytogenes* strains under different cold stress and bacterial growth media conditions have shown that this bacterium responds to low-temperature stress through translational and transcriptional bias that induces specific sets of proteins and transcripts [59,60,61,62,63,64]. Ultimately, these changes result in observable metabolome changes [65]. Induction of both Csp proteins and transcripts in response to cold stress exposure of *L. monocytogenes* has been detected [33,61,62]. CspA, one of the Csps that are strongly induced in response to cold stress, has also been found to be the most functionally important Csp for *L. monocytogenes* cold growth [33,62,66].

In *L. monocytogenes,* EGDe mutants devoid of *cspA* gene in the context of single (∆*cspA*), double (∆*cspAB* and ∆*cspAD*), and triple (∆*cspABD*), *csp* gene deletions are all not capable of growth at both 10 °C and 4 °C. However, CspA requirement for cold growth seems to be for relatively low temperatures since all the *csp* deletion mutants (single, double, and triple) of this strain were capable of growth until the stationary phase at 15 °C within 72 h [19]. Schmid et al. showed that at 4 °C *L. monocytogenes* variably upregulated its *csp* mRNA levels (*cspA* > *cspB* > *cspD*) compared to their levels at 37 °C [33]. These findings, combined with impaired growth phenotypes subsequently observed at low temperature for the *csp* deletion mutants, confirmed that Csps are involved in cold stress tolerance in this bacterium similar to what was observed in other bacterial species, such as *E. coli B. subtilis* and *Yersinia* spp [35,41,44,55,67]. Meanwhile, among the Csps, *cspA* determined to be the most important for growth at low temperatures was upregulated 5.1- and 3.1-fold compared to *cspB* and *cspD*, respectively, [33]. Interestingly, although *cspB* mRNA was upregulated more than *cspD* mRNA, the functional importance of CspB during cold growth was less than that of CspD. Strains lacking *cspB* were the least compromised in low-temperature growth with no observable growth defects in *cspB* deletion mutants when either *cspA* or *cspD* gene was present. On the other hand, in single and multiple *csp* deletion mutants, the loss of either *cspA* or *cspD* genes caused significant cold growth defects. Using a different strain, Hingston et al. observed that at 4 °C, *cspB* expression was repressed, while *cspA* and *cspD* were upregulated [66]. Overall, there is a clear hierarchical trend of CspA > CspD > CspB observed in Csp functional importance during low-temperature stress exposure of *L. monocytogenes*. Comparatively, deletion of single *csp* genes in *E. coli* and *B. subtilis* caused no distinct low-temperature growth phenotypes [68,69,70]. However, severe growth inhibition at both high and low temperatures was observed in *B. subtilis* upon deletion of two *csp* genes, whereas severe cold growth impairment only becomes apparent when at least four of the nine *E. coli csp* genes are deleted [71]. Such observations have thus pointed towards species-specific Csp roles and functional redundancies. Whether there might be genetic background-dependent differences, with Csps having varied roles depending on either the molecular subtype (genetic lineage, serotype, clonal complex) or strain of *L. monocytogenes,* remains to be investigated.

When exposed to cold stress, bacteria cell membrane fluidity and enzyme activity decrease. Moreover, the efficiency of transcription and translation is reduced due to the stabilization of nucleic acid secondary structures, while protein folding is inefficient and ribosome function is hampered [32,35,41]. In *E. coli*, Zhang et al. identified a Csp and exoribonuclease RNase R mediated mRNA structure surveillance system that facilitates translation recovery after cold shock [32]. It is postulated that at low temperatures, Csps function as chaperons to dynamically adjust or melt mRNA secondary structure, allowing translation to continue [32,41]. They are also thought to assist in refolding cold-stress damaged proteins and counteract the harmful effects of such stress on ribosome functions [32,41]. These chaperon activities might also be employed on negatively super-coiled DNA, ensuring DNA replication [35]. Interestingly, *L. monocytogenes* strains lacking *cspD* showed increased cold stress sensitivity in minimum defined media compared to rich media [33], either due to increased energy demands for cold tolerance when *cspD* is absent or due to some cryoprotective substances being present in the rich media that this mutant could utilize to withstand the cold stress and grow better.

It also appears that *L. monocytogenes* cold stress responses are not only important for survival in food but also for key virulence traits, such as cell invasion. Although cold stress exposure can either increase or lower *L. monocytogenes* host cell invasion capacity depending on the strain, more severe cell invasion attenuation effects were detected in *csp* deletion mutants [26,72]. In murine macrophages, the invasion capacity of *L. monocytogenes* EGDe wild-type, ∆*cspBD*, and ∆*cspABD* strains were 9-, 441-, and 1141-fold reduced, respectively, after cold stress exposure [26]. This is indicative that *L. monocytogenes* strains with reduced cold stress tolerance might be impaired in host cell invasion ability after exposure to cold temperature, probably due to surface protein changes and/or membrane damage post-cold exposure [26]. However, this needs to be investigated with analysis of more *L. monocytogenes* strains and *csp* mutants. Nonetheless, Csps seem to be crucial in maintaining cell invasiveness post-cold exposure, a situation most *L. monocytogenes* strains might encounter in foods before entering the host.

### 3.2. Osmotic Stress

*L. monocytogenes* efficiently adapts and sometimes proliferates, despite exposure to low temperatures, low pH, and elevated salt (NaCl) concentrations, conditions used in preserving RTE food products [26,33,64,73]. Osmotic stress is a fundamental preservative strategy used in foods with low water activity, such as salami, ham, bacon, and some deli meats, which periodically have been found contaminated with high numbers of *L. monocytogenes* [2,3,4,74]. Improvement of food safety measures taken against this pathogen will depend on further insights gained into molecular cell response mechanisms underlying osmotic stress resistance phenotypes displayed by this organism. The ability of *L. monocytogenes* to accumulate compatible solutes, such as glycine betaine and carnitine, has been implicated to be partly responsible for osmotic stress tolerance [16,64,75]. It has been suggested that they also function as cryoprotectants [16,65,76,77]. An *L. monocytogenes* EGDe mutant devoid of all *csp* genes could not grow when exposed to NaCl stress [33], revealing that Csps contribute to efficient cellular growth under NaCl salt stress. Analyzing various *csp* mutants revealed a hierarchical trend CspD > CspA/CspB in the relevance of Csps in NaCl osmotic stress resistance of *L. monocytogenes*. Strains lacking *cspD* were the most compromised in growth under osmotic stress, while those lacking *cspA* and *cspB* grew at rates analogous to the wild-type. However, double deletion of both *cspA* and *cspB* showed that these two genes also contribute to osmotolerance, albeit at lesser magnitudes than *cspD*. Interestingly, under the NaCl stress conditions applied in that study, only *cspD* and *cspA* genes were significantly upregulated at the mRNA level [33].

While mechanisms of Csp involvement in *L. monocytogenes* stress responses are yet to be elucidated, it is plausible that Csps might affect function or expression of the compatible solute transporter systems; glycine betaine porter I (BetL), glycine betaine porter II (Gbu), and the carnitine transporter (OpuC). Since CspA and CspD are the most critical for cold stress tolerance, these might contribute to the expression and regulation of these systems under cold stress, while CspD, which is more critical under osmotic stress, might be an effector under this stress. As CspD is an important player under both conditions, it might be responsible in part for the cross-protection observed against these two stresses in *L. monocytogenes*. Another possibility is that Csps might promote sodium ion extrusion transporter protein production, facilitating increased intracellular sodium extrusion and potassium accumulation, allowing growth under salt stress. Meanwhile, their chaperon activity might facilitate the repair of damaged DNA or altered DNA and RNA structures arising in osmotic stress exposed *L. monocytogenes* cells. The two-component regulatory system LisRK and the chaperone and serine protease high-temperature requirement A (HtrA) are known to play roles in osmosensing, osmoregulation, and osmotolerance in *L. monocytogenes* [21,78]. Csps might also exert their influences through these systems. Hence, it would be interesting to investigate the effects of *csp* deletion on these systems.

Recently, EGDe has been classified as one of the least osmotolerant *L. monocytogenes* strains [73]. In addition, lineage II strains to which EGDe belongs are also less osmotolerant than lineage I or III strains [73,79,80]. With this in mind, Csp roles in osmotic stress tolerance must be further validated in the context of these more osmotolerant strains or molecular subtypes.

### 3.3. Motility, Cell Aggregation, Biofilm Production, and Desiccation Tolerance

In the environment, motility is postulated to be important for chemotaxis towards nutrients and migration from stress, which might be key for survival and eventual contamination of food. Flagella also mediate surface attachment, inadvertently impacting biofilm formation [19,81]. Flagella are also known to act as pathogen-associated molecular patterns, which might be downregulated in part to avoid the immune system; however, some data suggest that motile *L. monocytogenes* perform better compared to nonmotile strains inside the host [82,83,84]. Involvement of Csps in motility has been reported in other bacteria, such as *Clostridium botulinum* [85]. Likewise, compared to their wild-type *L. monocytogenes* EGDe strain, single (Δ*cspA*) or multiple (Δ*cspAB*, Δ*cspAD*, and Δ*cspABD*) *csp* gene deletion mutants showed reduced or complete attenuation of swarming motility [19,28]. Conversely, ∆*cspBD* (expressing CspA) and ∆*cspD* retained wild-type level motility, while deletion of *cspB* alone had minimal effects on motility. Overall, these observations suggest that CspA is the most important Csp for motility expression as this phenotype was somewhat conserved when *cspA* was retained, while *cspD* seems dispensable for swarming motility. The importance of *cspA* in the expression of swarming motility phenotypes was also confirmed through ∆*cspA* mutants in different *L. monocytogenes* strain genetic backgrounds [19]. Csp roles in motility seem to follow the hierarchical trend CspA > CspB > CspD. While the data on multiple *csp* deletion mostly agrees among studies, differences are apparent in the finding with ∆*cspBD* (*cspA* only). In a study by Eshwar et al., this mutant, although capable of swarming motility, did so at a significantly reduced extent when compared to the wild-type strain, while according to Kragh et al., ∆*cspBD* retained wild-type level motility [19,28]. These differences are probably explained by the differences in experimental setup between these studies. For example, one was conducted on tryptone soya broth (TSB) agar at 15 °C over 8 days and the other on Brain Heart Infusion (BHI) agar at 25 °C over 2 days [19,28].

It appears these phenotypic deficits arise due to transcript-level-independent reduction of the effecter proteins of these phenotypes. Electron microscopic examination showed that, while the motile wild-type EGDe and *csp* deletion mutant strains (∆*cspBD* and ∆*cspAD* producing CspA or CspB only, respectively) exhibited peritrichous flagellation, there were no flagella observed on the surfaces of the nonmotile *csp* mutants ∆*cspABD* (without Csps) and ∆*cspAB* (producing CspD only). Overall, the flagellation frequency of the motile mutants was, however, relatively lower compared to the wild-type strain [28]. Inversely, compared to the wild-type, the *csp* deletion mutants had higher amounts of *flaA* transcripts, the gene that encodes flagellin, a major component of flagella. These observations indicate that Csps might exert their influence at the transcriptional, post-transcriptional, and translational levels ensuring efficient mRNA translation [28,32,35].

Regarding cell aggregation, the *csp* mutants demonstrated deficient phenotypes, with the ∆*cspAB* mutant producing CspB being the only mutant able to aggregate at a reduced extent compared to the wild-type. These *csp* mutants also had reduced or completely lacked ActA proteins but contained elevated *actA* mRNA levels compared to the parental wild-type strain supporting the idea that Csps are involved in post-transcriptional regulation of genes [28]. This observation and that of *flaA* suggest a broken positive feedback loop where the lack of the corresponding coded protein results in increased expression of the gene at the transcript level, although these transcripts are not efficiently translated to proteins. The cell aggregation phenotype linked to ActA deficiency might be key to virulence as ActA is critical for *L. monocytogenes* intracellular motility via polymerized actin, required for internalin C (InlC)-mediated cell-to-cell dissemination [86]. Moreover, these findings insinuate that Csps might be an important determinant of the expression of cell surface molecules, including ActA and flagella.

Cell aggregation, flagella, and extracellular matrix production are key factors in biofilm formation [87]. Since Csps are involved in the expression of these phenotypes, it would stand to reason that Csps might affect biofilm production. Similar to what was observed in bacteria, such as *Salmonella enterica* and *S. aureus* [54,88], analysis by Kragh et al. revealed that indeed Csps are involved in biofilm production, a phenotype important for persistence in the food production environment by increasing tolerance to heat as well as cleaning and disinfectant chemicals [16,19]. Using the peg lid assay approach, which simulates biofilm formation on surfaces where periodic cleaning occurs, washing away non-adherent cells or where changing water flows may add new nutrients, revealed that biofilm formation is greatly compromised in nonmotile *csp* deletion mutants (Δ*cspA*, Δ*cspAB*, and Δ*cspABD*) [19]. On the other hand, *csp* mutants (Δ*cspB*, Δ*cspD*, Δ*cspAD* and Δ*cspBD*), which still displayed some motility, produced biofilm analogous to their parental wild-type strain, highlighting the importance of motility for biofilm formation under these conditions. Surprisingly, using the traditional biofilm assay at 15 °C, all *L. monocytogenes* EGDe *csp* deletion mutants formed similar or more biofilm than their parental wild-type strain. In contrast, the opposite was observed for *L. monocytogenes* 568 and 08-5578 wild-type strains and their *cspA* deletion mutants [19]. These paradoxical observations indicate that *cspA* contributes to functions needed to produce biofilm in normal to high biofilm producers, while in poor biofilm producers, such as *L. monocytogenes* EGDe, *cspA* presence might be inhibitory to biofilm production. This also highlights a strain or molecular subtype associated trend in Csps functions since EGDe belongs to clonal complex CC9, while 568 and 08-5578 strains belong to CC8. Suggestions are that the increased biofilm formation is due to hyper-biofilm formation by the mutants due to failure of periodic cell shedding from the biofilm [19,81]. Moreover, this acts as a warning; in as much as treatments targeting Csps might impair motility, cold and/or osmotic stress tolerance. They might also inadvertently promote desiccation tolerance and biofilm formation, features key for long-term bacterial persistence in food environments.

Moisture in food processing environments promotes biofilm formation. To try and prevent this, manufacturers limit the availability of moisture in these environments [89]. However, some strains of *L. monocytogenes* can survive desiccation for extended periods on food contact surfaces [19,90]. Interestingly, nonmotile mutants with multiple *csp* gene deletions (Δ*cspABD* > ΔcspAB > Δ*cspAD* > Δ*cspBD*), or single (∆*cspA* > Δ*cspD* ≥ ΔcspB) exhibited elevated desiccation tolerance compared to their desiccation sensitive parent strain *L. monocytogenes* EGDe. More surprisingly, the triple *csp* deletion mutant Δ*cspABD* and the CspD expressing Δ*cspAB* mutant were the most tolerant to desiccation stress among the tested mutants [19]. Mechanistically, it is tempting to speculate that the increased tolerance in *csp* deletion mutants shows that Csps might have a negative regulatory role on systems employed to ensure desiccation tolerance. The upregulation of desiccation-related genes *qoxB* and *pdhA* observed in *csp* deletion mutants in part supports this notion [19]. Alternatively, these findings might be indicative of dysregulation of these systems in the absence of Csps. These tolerance phenotypes could also be related to loss or reduction of motility in these mutants. Other studies have suggested that decreased or absence of motility might favor enhanced desiccation tolerance as enormous cellular resources are expanded to build and run the motility apparatus [91,92]. However, deletion of *cspA* only in the more desiccation-resistant food and outbreak-related *L. monocytogenes* strains (568 and 08-5578) had no impact on desiccation tolerance, although compared to the parent strains, the mutants were also nonmotile [19]. Expression of *cspA* was upregulated in wild-type EGDe, 568 and 08-5578 strains in response to desiccation stress confirming a role for *cspA* during desiccation survival [19,92]. Therefore, the fact that *cspA* deletion had no impact on 568 and 08-5578 strains desiccation stress sensitivity is indicative that functional redundancy might exist in these two strains regarding desiccation, which might not exist in EGDe. In support of this, strain 568 and 08-5578 Δ*cspA* mutants expressed significantly elevated levels of *cspD* and *cspB* genes (>30-fold higher) compared to their wild-type strains [19]. This upregulation probably compensated for the deleted *cspA* gene, a phenomenon also observed in other species like *S. enterica*, *E. coli*, and *B. subtilis* [36,54,55,68,71]. Taken together, these observations show that some *csp* functional roles and redundancies are conserved, while some vary among *L. monocytogenes* strains with different genetic backgrounds on some phenotypes.

Overall, the observations suggest molecular subtype-associated functional redundancies and functional differences among the three *L. monocytogenes* Csps in their contributions to cellular aggregation, surface flagellation, swarming motility, desiccation tolerance, and biofilm production.

### 3.4. Csp Involvement in Acid Stress Response, Metabolism, Hydrostatic Pressure, and Antimicrobial Resistance

In the journey to the host cell, *L. monocytogenes* is confronted with pH stress in acidic foods, during gastric passage and in the macrophage phagosomes [93]. The pathogen responds to and survives in these low-pH environments by utilizing several stress adaptation mechanisms [16]. Csps have been implicated in pH stress response in other bacteria, including *C. botulinum* [41,85]. However, their involvement in *L. monocytogenes* pH stress tolerance is yet to be investigated. Similar to observations in *E. coli*, *L. monocytogenes* Csps were induced in response to high hydrostatic pressure (HHP) treatment, signaling their potential involvement in adaptation to pressure treatment [61]. Since Csps are induced under cold and osmotic stress, these observations imply that cold or osmotic stressed bacteria could be more tolerant to HHP stress, which might affect the efficiency of such combined food preservation techniques due to Csp-mediated cross-protection.

Czapski and Trun, using *E. coli,* demonstrated that different Csps are important at optimal temperature in different media [56]. In addition, some studies have shown differences in Csp abundances at different stages of the growth cycle of a culture [94]. No significant growth phenotype differences were observed between the wild-type *L. monocytogenes* EGDe strain and its corresponding single, double, and triple *csp* deletion mutants at 37 °C in rich (BHI) and minimal defined media conditions [28,33]. Despite *csp* deletion mutants growing at the same rate as the wild-type at 37 °C or growing and reaching stationary phase at 15 °C, their metabolic activity or capacity to utilize certain nutrients or carbon sources might be compromised. In support of this, during desiccation, mRNA levels of pyruvate dehydrogenase (*pdhA*), which links glycolysis and the incomplete *Listeria* citric acid cycle [95,96], were significantly lower in the *csp* mutants compared to their parent strains [19]. In *Brucella melitensis* and *S. aureus,* differential expression of metabolism-associated genes was observed when *cspA* was deleted [30,38]. This suggests that *cspA* of both bacteria plays a role in metabolism regulation. Analysis of *csp* deletion consequences on metabolism or nutrient utilization of *L. monocytogenes* is, therefore, warranted as its Csps might have analogous effects on metabolism.

Csps influence the expression of several phenotypes in other bacterial species, such as ethanol stress tolerance in *C. botulinum* ATCC3502, buffering of deleterious mutations, and persister cell formation [41,85]. It would be interesting in the future to also investigate if Csps influences responses to these stresses in *L. monocytogenes*. Nisin is an important bacteriocin applied to foods targeting *L. monocytogenes*; Wu et al. observed downregulation of *cspB* in *L. monocytogenes* NZRM 4734 persister cells exposed to lethal nisin levels (75 µg/mL) [97]. Conversely, Liu et al. observed induction of *cspA* and *cspD* in *L. monocytogenes* Scott A exposed to nisin at a sublethal dose of 20 µg/mL [98]. This is suggestive that Csps might have functional roles in *L. monocytogenes* response to nisin stress. It would be prudent to further investigate their influence on nisin tolerance, especially at nisin concentrations analogous to those applied in foods.

Although *L. monocytogenes* is not well-known for its antimicrobial resistance (AMR), the latter is an ever-present threat to public health [16]. Understanding all mechanisms that might be involved in resistance is key in the fight against AMR. In *E. coli,* it has been reported that Csps can contribute to the bypass of the inhibitory effects of some antibiotics [99]. Cruz-Loya et al., using a library of transcriptional fluorescent reporters, confirmed that cold-shock genes are induced by antibiotics in *E. coli* [100]. They postulate that elements of the cold and heat stress responses have evolved to be deployed against multiple classes of antimicrobials. In *L. monocytogenes*, *cspA* was induced in response to exposure to sublethal concentrations of the natural antimicrobial compounds citral and carvacrol [101]. Since Csps are involved in various stress responses and affect systems that are also targeted by different antibiotics, it would be prudent to evaluate their roles in *L. monocytogenes* antibiotic stress response. In addition to antibiotic resistance, the emergence and spread of resistance to cleaning and disinfectant chemicals in *L. monocytogenes* are greatly concerning [16,102]. It is, therefore, important that we establish Csp roles in responses to such stressors.

## 4. Role of Csps in *L. monocytogenes* Virulence

The transition from saprophyte to an intracellular pathogen of *L. monocytogenes* hinges on it implementing a wide range of well synchronized molecular events designed to resist host defenses and facilitate the infection of different target host cells. These events are coordinated through complex gene expression regulatory networks, including but not limited to the central virulence regulating protein PrfA, σ^B^, and the two-component-based ViR/VirS virulence regulatory protein system [22,57]. Transcriptome analysis revealed that *cspA* and *cspB* genes were among other *L. monocytogenes* genes upregulated during host infection, indicative that they might play a role in the expression of its virulence traits [103]. Indeed, their involvement was confirmed both in vitro (cell culture) and in vivo (zebrafish embryo microinjection), where *L. monocytogenes* lacking *csp* genes were severely impaired in epithelial cell invasion and murine macrophage infections as well as zebrafish embryo pathogenicity [26,27,28]. Interestingly, the ∆*cspBD* and ∆*cspABD* strains are not impaired in adherence to the Caco-2 cell surfaces. Their adhesion indexes are higher than the wild-type, pointing towards reduced internalization or survival in the host cells as the reasons for their impaired virulence. It would be prudent to analyze transcript and protein levels of InlA and InlB in these *csp* mutants, as these two surface proteins are critical for entry into non-phagocytic host cells. At present, the Csp controlled cellular mechanisms that might facilitate *L. monocytogenes* virulence are not yet fully understood. However, emerging data highlight Csps roles in promoting intracellular stress survival of *L. monocytogenes* during host cell infection and expression of key virulence factors.

Within the host, bacteria encounter stress, such as oxidative stress. Overcoming such stress and escaping the phagocytic vacuole to gain access to the nutrient-rich cytoplasm and replicate is key to infection establishment. Csps are postulated to play a critical role in facilitating oxidative stress tolerance, probably by regulating the expression of oxidative stress response machinery [26]. Reduced oxidative stress tolerance noted among some *csp* deletion mutants (Δ*cspBD* and Δ*cspABD*) might be one reason for their reduced survival and growth in murine macrophages [26]. The reduced growth in macrophages might also highlight deficiencies in nutrient utilization intracellularly by these *csp* deletion mutants. Studies to confirm this hypothesis are, however, required. In addition to promoting antioxidative responses, it has been confirmed that Csps influence expression of *hly,* which codes for the pore-forming cytolysin LLO in a PrfA-independent manner [27]. Transcripts of the *hly* gene were lower in the absence of all three *csp* genes or *cspB* alone, corresponding to reduced hemolysis in these mutants. In part, this reduced transcript level was due to decreased *hly* mRNA stability in *csp* gene deletion mutants, suggesting that Csps are involved in mRNA turnover events [27]. In another study, Eshwar et al. detected reduced levels of proteins or activity as well as transcripts from the *Listeria* pathogenic island-1 (LIPI-1) genes *prfA*, *hly*, *mpl*, and *plcA* in mutants lacking two (∆*cspAB*, ∆*cspAD*, ∆*cspBD*) or all three *csp* genes (∆*cspABD*) suggesting a Csp-dependent transcriptional and post-transcriptional regulation of these genes and their coded proteins. However, an exception to such trends was observed for *hly* and *mpl* mRNA levels, which were not significantly different from the wild-type strain in RNA derived from BHI cultured bacteria of the *cspB* harboring ∆*cspAD* mutant [28]. LLO, phospholipases PlcA and PlcB, and the metalloprotease Mpl facilitate *L. monocytogenes* escape from vacuoles or phagosomes into the host cell cytosol where the bacteria replicate [103,104]. The reduced *hly*, *mpl*, and *plcA* transcript levels and their corresponding proteins, as well as attenuated ActA production, might further explain the reduced survival and growth of *csp* deletion mutants within host cells.

Among the *csp* mutants, strains producing only CspD had the greatest PrfA and LLO protein levels, followed by strains producing CspB, whereas CspA producing strains consistently showed the lowest amounts for these proteins. On the other hand, Mpl levels were similar to slightly different between the strains producing only CspA or CspB. However, compared to strains producing CspD only (∆*cspAB*), these levels were significantly lower [28]. Interestingly, although PrfA and LLO protein levels in ∆*cspAD* were lower than the ∆*cspAB* strain, ∆*cspAD* contained higher *prfA*, *hly*, and *mpl* mRNA amounts and was significantly more hemolytic and virulent in zebrafish embryos than the other *csp* deletion mutants [28]. This indicates that Csps might be involved in pre- and post-translational events and the mechanisms through which Csps exert their influences are more complicated than initially thought.

Overall functional redundancy, hierarchical trends, and segregation of duties are evident regarding Csp roles in virulence. For instance, cells devoid of CspA had no significant difference in Caco-2 cell invasion compared to the wild-type, while those carrying single deletions of *cspB* and *cspD* or double and triple *csp* deletions (∆*cspAB*, ∆*cspAD*, ∆*cspBD*, and ∆*cspABD*) had significant differences to the wild-type strain. Indicating that if one or both *cspB* and *cspD* remain intact, the loss of *cspA* function has minimal consequences on *L. monocytogenes* cell invasion phenotypes. The Caco-2 and macrophage cell invasions displayed between ∆*cspB* and ∆*cspAB*, and those between ∆*cspD* and ∆*cspAD* strain pairs, were comparable [26]. Interestingly, ∆*cspBD* and ∆*cspABD* strains showed the most severe reduction in cell invasion capacity, more so in the triple deletion, while intramacrophage survival and growth rates were more significantly impaired in ∆*cspABD* strains highlighting that CspA somewhat contributes to some aspects of host cell invasion [26]. In addition, cells of strains devoid of all *csp* genes could not be detected in THP-1 macrophages 24 h post-infection, while the wild-type strains and double *csp* deletion mutant (∆*cspAB*, ∆*cspAD*, ∆*cspBD*) cells increased to varying levels [28]. Furthermore, single *csp* gene deletion did not impact hemolysis on blood agar plates, while significant differences were observed with double (∆*cspBD* or ∆*cspAB*) and triple *csp* deletion mutants [27]. Unlike the situation for hemolysis where CspB could restore wild-type level phenotypes, no single *csp* expressing strain had cell invasion or intramacrophage survival and growth rates similar to the wild-type. This suggests that functional activities of all three *csp* genes are necessary for maximal cell invasion, survival, and growth of this bacterium during human epithelial and macrophage cell infection. Taken together, these observations further confirm Csps roles in virulence, highlighting functional bias and hierarchical trends in their influence on virulence-associated phenotypes.

In zebrafish embryos infected with wild-type and *csp* mutants of *L. monocytogenes* EGDe strain, variable mortality induction was observed, with all *csp* deletion mutants showing lower virulence than the wild-type [28]. These *csp* deletion mutants took longer (24 h later) to cause 100% mortality of the zebrafish embryos compared to the wild-type strain. In line with the in vitro findings, no ∆*cspABD*-infected embryos succumbed to the infection throughout the trial [26,28]. Moreover, CFU counts tracking the growth of *csp* deletion mutants in zebrafish embryos showed a reduced growth rate in the triple *csp* deletion mutant, with its bacterial numbers even decreasing towards the end of the trial [28]. Data generated using the zebrafish embryo infection model provides valuable insights into the roles of Csps in virulence. However, it has some drawbacks. The optimum temperature for this experiment is 28 °C, a temperature lower than the human core body temperature. At this temperature, Csps might exert a greater influence. Furthermore, this model bypasses the gastrointestinal phase of the infection process, which is critical since listeriosis mainly arises after ingestion of *L. monocytogenes* contaminated food. It would be prudent to further validate these findings in a system that reflects better the situation in humans in terms of temperature and other host factors. Combining both the in vitro and in vivo findings from these studies, a general hierarchical trend CspB > CspD > CspA emerged in terms of virulence-associated phenotypes [26,27,28]. Interestingly, under optimal growth conditions in BHI at 37 °C, *cspB* and *cspD* transcripts were 20- and 4-fold, respectively, higher in abundance than *cspA* mRNA in *L. monocytogenes* [33].

The full extent of Csps influences on virulence remains to be investigated. Analysis of the effects of single *csp* gene deletion on virulence-associated phenotypes in in vivo settings has not yet been done. However, it is expected that the effects of functional redundancy will be more apparent.

## 5. Csps Regulation

In a departure from their name, Csps are not only cold-induced but are expressed under various conditions. As already detailed, they are important for several cellular processes that range from stress tolerance to virulence (Figure 2). This feature is not unique to *L. monocytogenes* as it is also seen in other bacterial species, such as *Brucella melitensis*, *B. subtilis*, and *E. coli,* to mention a few where *csp* genes are downregulated or upregulated in response to stress or during infection [38,41,61]. Generally, not much is known about *csp* regulation in *L. monocytogenes*. In *E. coli, cspA* is regulated at the transcription, post transcription, translation, and post-translation levels [44]. Zhang et al. proposed the presence of an autoregulatory switch that allows Csps to remodel their own 5’ untranslated leader regions (5’ UTR), fine-tuning their own expression based on cellular needs enabling dynamic regulation of overall translation [32]. They also concluded that the widespread distribution of highly conserved Csps in bacteria alludes to broad utilization of these control mechanisms. On this basis, it is expected that similar regulatory structures exist for *L. monocytogenes csp* genes.

Similar to the situation in other bacterial species [43,44,68,105,106], *L. monocytogenes csp* protein-coding mRNA regions are preceded by long 5’ UTR initially postulated to range from 198 bp on *cspA* to 363 bp on *cspD* [33]. The 5’ UTR lengths were later experimentally confirmed through RNA sequencing to be *cspA* (110 bp), *cspB* (102 bp), and *cspD* (124 bp) [23]. These relatively long 5’ UTR regions might have implications in the regulation of *csp* gene expression in this organism. These regions were predicted to have hairpin structures which might affect translation efficiency and conditions under which translation occurs. In *E. coli*, an AT-rich sequence upstream the -35 region of the *cspA* promoter enhances the transcription of this gene at low temperature [107]. Similar hairpin structures are known to affect the translation of some genes in *L. monocytogenes.* A well-known example is the 5’ UTR thermosensor of *prfA* mRNA, which folds into a hairpin structure at temperatures below 30 °C to inhibit translation initiation [108,109]. Interestingly, in *E. coli,* the *cspA* mRNA long 5’UTR also functions as a thermosensor that modulates translation of the cold shock protein CspA in this organism [32,110]. Recently, Ignatov et al. demonstrated that an RNA thermoswitch also controls the expression of CspA in *L. monocytogenes* [111]. The mRNA of *cspA* senses temperature changes and adapts to these changes by assuming a conformation that allows for increased translation efficiency under cold stress [110,111]. However, in *E. coli,* it does so by adopting functionally distinct structures at different temperatures, which do not necessarily result in the melting of hairpin structures [110], while in *L. monocytogenes,* this is through disruption of these hairpin structures exposing the Shine–Dalgarno sequence, allowing ribosome binding [111]. These structural forms also increase the mRNA stability decreasing their degradation compared to the *cspA* mRNA structure at 37 °C [105,107,110]. This type of regulation is not unique to *cspA* only as translation of *cspB,* and *cspC* of *S. aureus* is also controlled by RNA thermoswitches [112]. In addition, Csp proteins are generally stable from degradation even under stress, presumably ensuring that they have an extended half-life to exert their influence [113]. Since similar regulatory mechanisms have also been observed in several other bacteria, such as *Thermus thermophilus* [41,114], such regulatory mechanisms might be widespread among bacterial species, including *L. monocytogenes*.

*Yersinia* spp. employs a unique system as part of its *csp* gene expression regulation mechanism, *cspA* mRNA can be produced as both monocistronic and/or bicistronic mRNA templates [115]. The favored transcript produced is determined by the temperature stress the cell is exposed to, with the monocistronic predominating at high temperatures, while under acute cold stress, bicistronic mRNA is produced, shifting to monocistronic mRNA as cold stress exposure is prolonged [115]. The inherent advantage is that translation is more efficient when the transcript contains two copies of the protein giving these organisms a better adaptive advantage [115]. Interestingly, a similar system exists in *L. monocytogenes* regarding *prfA* transcriptional regulation. Two of its promoters, P*_prfa1_* and P_prfa*2,*_ are located immediately upstream of *prfA* and generate initial levels of PrfA protein via monocistronic mRNA. A third, bicistronic transcript originates from read-through off the promoter of *plcA* upstream of *prfA* [22,104,116]. This third promoter is PrfA-dependent and thus results in an autoregulatory feedback loop. It would be interesting to check if there are similar mRNA-based regulatory mechanisms present in *L. monocytogenes* concerning Csp regulation.

Csps might also have a negative or positive regulatory loop feedback system where overexpression or downregulation of one *csp* affects the expression levels of another. In support of this, Kragh et al. observed what they proposed to be a compensatory increase in *cspD* and *cspB* transcripts in *cspA* deletion mutants compared to their wild-type in response to desiccation [19]. This feature was also noticed in other bacterial species [41,55,68], indicative that Csps might regulate themselves and each other’s expression, possibly in a negative feedback loop manner as observed in other bacteria such as *S. aureus* for CspA or CspE regulating *cspA* expression in *E. coli* [30,44,112,117]. A triple *csp* deletion (∆*cspABE*) in *Lactococcus lactis* did not affect growth characteristics of the bacterium, an observation attributed to increased expression of the remaining *csp* genes [44]. This functional redundancy means that a phenotype might only be observed when a significant number or all *csp* genes of a bacterium are deleted. In addition, emerging data suggest that *cspD* is regulated by alternate sigma factors. It is known that under osmotic stress, σ^B^ negatively regulates *cspD* while this function is assumed by σ^L^ in late logarithmic phase cultures [22,118,119]. Conversely, the class I stress response gene regulator HrcA, which is primarily a negative regulator, indirectly positively regulates the expression of *cspD* [22,120]. It seems an intricate system of different transcriptional regulators and mechanisms exists to fine-tune *csp* gene expression under various conditions. Since Csp roles have a strain-associated trend under some conditions, some strain and/or phylogenetic diversity in *L. monocytogenes csp* gene expression regulation is, therefore, expected.

## 6. Future Perspectives

A growing body of data supports bacterial Csp involvement in transcriptional and translational regulation of numerous genes to promote a diverse set of traits, including those critical for the public health and food safety impacts of pathogens [38,54,88]. It has been observed that bacterial Csps are involved in the execution of cellular functions such as normal bacterial growth, adaptation to nutrient starvation, and stationary-phase growth [32,121]. Moreover, based on studies in several microorganisms, Csp family proteins have also been linked to modulation of microbial adaptive responses to antibiotics, elevated osmolarity, ionic compounds, and oxidative stress conditions [30,33,41]. As observed in other microbes, this influence seems to be exerted through increases in mRNA levels of effecter genes either via promoting their transcription or increasing the mRNA stability [27,28,47,49,50,54,105]. In *E. coli,* this function is mediated by CspE, which binds to RNA and impedes poly-(A)-mediated 3’-5’ mRNA decay due to polynucleotide phosphorylase [49]. Apart from increasing transcript levels, Csps can increase protein synthesis through direct interactions with transcripts leading to translation promotion either by facilitating translation initiation or through destabilization of translation inhibiting mRNA secondary structures [47,105]. Csps can also regulate genes via indirect mechanisms by affecting the regulators of these genes. Csp-dependent regulation of PrfA expression supports this as *L. monocytogenes* LIPI-1 genes that are under PrfA control were also differentially expressed in *csp* absence [28].

Csp deletion in the *L. monocytogenes* reference strain EGDe has indicated the importance of Csps in stress resistance and virulence functions in this bacterium, a feature also observed in other bacteria, such as in *B. melitensis* and *S. enterica* [38,54]. Deletion of *csp* has varied effects on different bacterial species. For example, deleting all three *csps* genes in *B. subtilis* results in a lethal phenotype, while deleting all *csps* in *L. monocytogenes* does not [33,55]. Csps seem to have functional redundancy as more than one Csp affects certain phenotypes albeit, at different extents. For instance, all Csps are somewhat involved in low temperature and osmotic stress tolerance, although CspA and CspD are the most important for each stress, respectively. These functional redundancies among the three Csps of *L. monocytogenes* are also observed in phenotypes, such as hemolysis, virulence, flagella-based motility, osmotic and cold stress tolerance (Table 1). The extent of this redundancy is, however, varied with the phenotype in question. For example, none of the single-gene deletion mutants of each of the three *csp* genes caused observable phenotypic defects in hemolysis [27]. Based on this, it can be assumed that the impacts of individual *csp* genes deletion on LLO production are limited due to functional redundancies. Molecular function redundancy between Csps is expected due to their high nucleotide and amino acid sequence conservation [33]. Furthermore, it seems each Csp might have different targets or roles in post-transcriptional gene expression regulation. For instance, for LLO production, although *hly* mRNA detected in Δ*cspAB* was lower than in Δ*cspBD*, supernatants from this strain contained higher LLO concentrations than those detected in Δ*cspBD* [27]. Interestingly, for most phenotypes that have been examined thus far, no single *csp* expressing strains could recapitulate wild-type level phenotypes except *cspB* on hemolysis [27]. The same is noticed in other bacterial species, and we postulate it to be a division of labor and a backup system to ensure that important phenotypes are preserved. For example, in *S. enterica*, double deletion of *cspC* and *cspE* (∆*cspCE*) resulted in strong virulence or stress tolerance deficiency phenotypes, whereas single gene deletions had limited to no consequences, signifying CspC and CspE functional redundancy [54].

Different functional roles and phenotypes were observed regarding desiccation and biofilm production when Csps were deleted from strains representing different genetic backgrounds. Inversely, some functions were maintained. For instance, *cspA* deletion resulted in reduced motility in all strains regardless of their molecular subtype or genetic background. This highlights Csp functional role conservation and diversity among strains from different genetic backgrounds on selected phenotypes. Thus, these findings validate the need for future analysis of Csp roles in strains from different genetic backgrounds, especially those of greater public health and food safety importance.

Cross-protection between stressors has been reported for many hurdle techniques against *L. monocytogenes* [9,122,123,124,125]. A good example is cross-tolerance observed with osmotic and cold stress or cross-protection to lethal oxidative and alkali stress upon exposure to cold stress [16,26,124,125]. This might be due to the induction of common genes and proteins essential to resist these stresses in *L. monocytogenes*. Some of the measures used for cold stress tolerance, like the accumulation of compatible solutes, are employed against osmotic and cold stress. Since Csps are involved in tolerance to many of these stressors, they might be the link for this cross-protection. The fact that Csps promote *L. monocytogenes* adaptation against both cold and NaCl stress has significant implications in view of practical food microbial control measures. The combined or sequential exposure of *L. monocytogenes* cells to these two stresses in food environments might inadvertently induce cross-protection responses.

Besides increased cold and salt stress sensitivity, an *L. monocytogenes* EGDe *csp* null mutant is severely impaired in virulence, oxidative stress tolerance, cell aggregation, and motility. Since these attributes are central to the current public health and food safety problems associated with *L. monocytogenes*, we must understand the role of Csps in these phenotypes. A major caveat is, however, that most experiments to investigate *L. monocytogenes* Csp roles have so far been examined in a few strains belonging to CC8 and CC9. *L. monocytogenes* EGDe, which has been used to generate the bulk of knowledge on Csp roles in this organism, represents a molecular subtype rarely implicated in human clinical illnesses [34]. It is, therefore, important that Csp roles are further examined in the context of more clinically important and highly virulent *L. monocytogenes* genetic backgrounds. An improved understanding of Csp involvement in *L. monocytogenes* cellular functions will require identifying and characterizing the Csp regulated genes, which has not yet been done in this pathogen. It is important that in the future Csp-dependent stress response and virulence phenomes, as well as their regulons, be determined, taking into account the genetic and phenotypic diversity of the more clinically relevant *L. monocytogenes* strains.

Although interesting roles and possible functions of Csps have been reported, at this stage, most of the functional roles and mechanisms of actions of Csps in *L. monocytogenes* are still based on speculation or extrapolations based on knowledge derived from the model organisms, such as *E. coli* and *B. subtilis*. Therefore, these remain to be validated in the context of this bacterium. The referenced studies have provided a basis for future work on Csp roles in *L. monocytogenes,* validating the need for in-depth analysis into stress and virulence-promoting processes modulated through Csp-dependent mechanisms in this bacterium. Pertinent questions, such as, “Do Csps interact with 5′-UTR of mRNA as observed for *S. aureus* CspA [30,112], do they unmask mRNA ribosomal binding sites, if so, by which mechanism, and do they interfere with the interaction of coding and regulatory RNA?”, remain unanswered.

Csps are induced in several stress conditions, and such induction under one stressor might be involved in cross-protection phenotypes against other stress factors, such as acid, osmotic, and cold stress. The cross-resistance and cross induction might also be critical to virulence as Csps were shown to be important for full virulence of *L. monocytogenes*. This has important implications for food safety applications, as interventions intended for improved food safety might increase virulence. In addition, this might be important for the design and application order of hurdle technique combinations in the fight against *Listeria* in foods. Therefore, a full definition of the global phenotypic and regulatory roles of Csps in *L. monocytogenes* is critical. Such data can be used to optimize anti-*Listeria* hurdle procedures and create new strategies and interventions to control this pathogen, thus improving food safety.

## Figures and Tables

**Figure 1 microorganisms-09-01061-f001:**
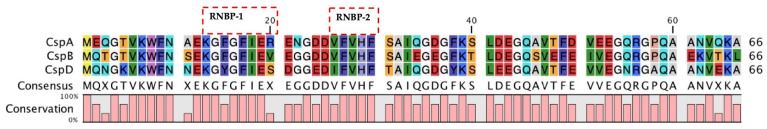
Csp protein sequences based on *L. monocytogenes* EGDe genome, highlighted under the red rectangles, are the two RNA-binding motifs (RNBP). Figure created using QIAGEN CLC Genomics Workbench 20.0 (available online: https://digitalinsights.qiagen.com/ (accessed on 9 April 2021)).

**Figure 2 microorganisms-09-01061-f002:**
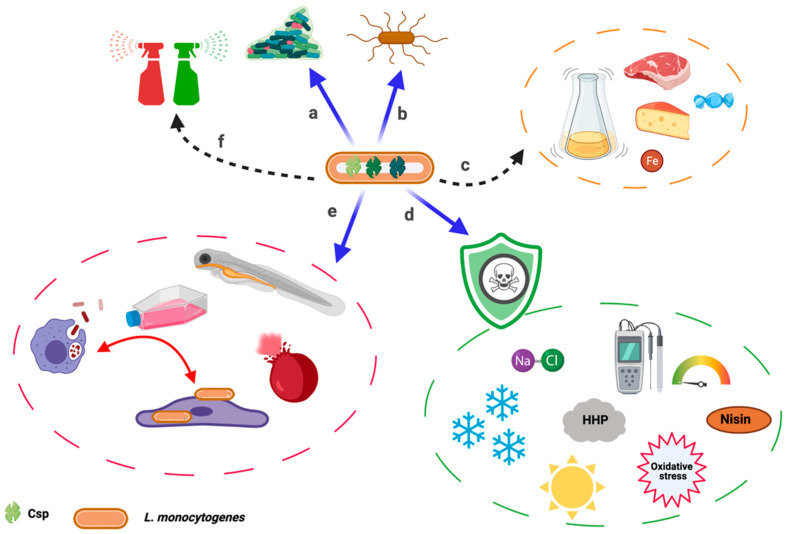
Overall effects of cold shock proteins (Csps) in *L. monocytogenes*: Csps are involved in (**a**) biofilm formation, (**b**) motility through impacting flagella expression, (**c**) they are dispensable during normal growth at optimal temperature. However, current data and inference from other bacterial species suggest that Csps are involved in nutrient acquisition and utilization. Nevertheless, this needs to be confirmed. (**d**) Csps are involved in stress tolerance responses to cold, osmotic, oxidative, high hydrostatic pressure (HHP), and desiccation stress. It is suggested that they might play roles in response to pH and bacteriocin (e.g., nisin) stress. However, this needs to be validated. (**e**) Csps are important for the full expression of virulence as observed in zebrafish embryos. They promote hemolysis, cell invasion, as well as growth and survival within macrophages. (**f**) Csps involvement in response to cleaning and disinfection chemical stress is yet to be confirmed, but inference from other bacterial species supports their involvement. Figure 2 was created with BioRender.com.

**Table 1 microorganisms-09-01061-t001:** Hierarchical trends of Csps influence on selected phenotypes in *L. monocytogenes.*

Phenotype	CspA	CspB	CspD
Cold stress tolerance			
Osmotic stress tolerance			
Oxidative stress ^a^			
Desiccation tolerance			
Nisin			
Hydrostatic pressure			
Biofilm			
Motility			
Zebrafish embryo virulence			
Hemolysis			
Epithelial cell invasion			
Macrophage cell invasion			
Intracellular growth			

**^a^** Effect only noted in double deletions of *cspB* and *cspD*; color codes define hierarchical influences each protein has on each phenotype: red > orange > yellow.

## Data Availability

Not applicable.

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
