# Peer review of "Listeria monocytogenes Cold Shock Proteins: Small Proteins with A Huge Impact"

_microorganisms, 2021, doi:10.3390/microorganisms9051061_

Round 1

Reviewer 1 Report

The present review manuscript is overall well constructed and written, providing a useful compilation of published scientific knowledge regarding the role of cold-shock proteins in various stress-related responses of the important foodborne pathogen Listeria monocytogenes

There only some minor comments regarding the submitted manuscript prior to acceptance of publication of the manuscript, and which are outlined below:

1. A general comment is the excessive length of many sentences throughout the manuscript, making them hard to follow and comprehend. Please add commas (,) or semicolons (;) for simplification, and when needed also proceed with splitting very long sentences to at least two distinct ones separated with periods (.). Some indicative examples of such long sentences are the following:

  • L126-128: "Csps have traditionally been known...in L. monocytogenes"
  • L195-197: "Whether there might be genetic background-dependent differences...strain of L. monocytogenes remains to be investigated"
  • L204-206: "It is postulated that at low temperatures Csps...effects of such stress on ribosome functions"
  • L220-223: "This is suggestive that L. monocytogenes strains...more L. monocytogenes strains and csp mutants"
  • L265-267: "Recently, EGDe has been classified...lineage I or III strains"
  • L344-346: "Moreover, this saves as a warning that...persistence in food environments."
  • L449-451: "Interestingly, the ΔcspBD and...as the reasons for their impaired virulence."
  • L541-544: "Analysis of the effects of single csp gene deletion...will be more apparent."
  • L650-653: "Csps seem to have functional redundancy...are the most important for each stress, respectively"

2. L10: I would suggest changing the sentence "This has cemented it as a significant food safety and public health issue." to "For this reason, L. monocytogenes has been identified as a significant food safety and public health concern"

3. L29: please correct "bacteria" to "bacterium"

4. L57-58: please revise to "In order for this "lifestyle" to be maintained, effective sensing for the distinction between the saprophyte and intercellular states is needed"

5. L71-72: please consider revising to the following for clarification purposes: "These attributes are central to the food safety and public health issues associated with L. monocytogenes"

6. L73-74: revise to "An existing limitation is that the strain EGDe, so far employed in Csp studies, represents a L. monocytogenes genotype rarely involved in clinical human illnesses"

7. L91-93: the sentence "L. monocytogenes Csps similar to those described in other bacteria...which are conserved between different genetic backgrounds in this bacterium" should be revised for syntax

8. L134: correct to "bacteriocins"

9. L145: please change "suggests" to "suggesting"

10. L154: please replace comma (,) with a semicolon (;) between "strategy" and "low"

11. L157: please revise to "With L. monocytogenes being a psychrotroph, its ability to..."

12. L160: correct to "in refrigerated foods"

13. L167-168: revise to "CspA, one of the Csps that are strongly induced in response to cold stress, has also been found..."

14. L182-183: please revise to "as the growth of strains lacking this gene was completely inhibited at low temperatures"

15. L215: please revise to "...survival in food but also for cell invasion, a key virulence trait"

16. L215-217: revise to "Although cold stress exposure can either increase or lower L. monocytogenes host cell invasion capacity depending on the strain, the cell invasion attenuation effects were found to be more severe in csp deletion mutants"

17. L220 and wherever else applicable throughout the text of the manuscript: use "indicative" in the place of "suggestive"

18. L222: please correct to "and/or"

19. L225: please add a comma (,) after exposure

20. L226: change to "in foods"

21. L231: I assume that you mean "ham" and "bacon"

22. L238: what do you mean by "double"?

23. L240: please change to "NaCL stress"

24. L243: add a comma (,) between "stress" and "whilst"; similarly in L253

25. L249: please add a comma (,) after "to be elucidated"

26. L251: change to "more critical" or "the most critical"

27. L256: either use "Csp proteins" or "Csps" (as in L263)

28.  L288: use "among: instead of "between"

29. L291: since you have not used the WT acronym so far, better use "wild-type strain"

30. L321: use "key factors" or "key parameters"

31. L375: use directly "S. enterica" (full name already mentioned before in the manuscript"; similarly in L669

32. L376: please correct to "...these data show"; singular: datum, plural: data

33. L379: please revise to "Overall, the data are indicative..."

34. L412 and wherever else applicable in the manuscript: correct to "bacterial species"

35. L416: replace comma (,) with a semicolon (;) after "L. monocytogenes"

36. L414: delete comma (,) after "cell formation"

37. L422: correct to "...analogous to those applied in foods"

38. L423-424: please revise to "Although L. monocytogenes is not well known for its antimicrobial resistance (AMR), the latter is an ever-present threat to public health"

39. L426: please revise to "Csps can contribute to the bypass of the inhibitory effects of some antibiotics"

40. L429: better to use the more general "antimicrobials" instead of "antibiotics"

41. L446: delete "in" prior to "in vitro"

42. L456: correct to "emerging data highlight..."

43. L459: revise to "...oxidative stress, and overcoming such stress..."

44. L496: revise to "more complicated than initially thought"

45. L519: add a comma (,) after "virulence"

46. L523-524: please revise this sentence for clarification

47. L528-530: please revise this sentence for clarification

48. L530: use "mainly" instead of "chiefly"

49. L535: use "reflects better" in the place of "more reflects"

50. L553: correct "level" to "levels"

51. L572: what do you mean by "doubles as a thermosensor"?

52. L575-576: please revise this sentence for clarification

53. L616: correct to "data are indicative"

54. Table 1: correct to "Epithelial cell invasion"; also in the footnote change "that phenotype" to "each phenotype"

55. L690, 727-728: there is also the likelihood for cross-protection between acid and heat tolerance...any available information regarding this?

56. L727-729: please revise to "Csps are induced in several stress conditions, and such induction under one stressor might be involved in cross-protection phenotypes against other stress factors such as acid, osmotic and cold stress"

57. L733: correct to "in foods"

Reviewer 2 Report

General assessment and comments:

In the submitted manuscript, Muchaamba et al. reviewed current knowledge of Listeria monocytogenes cold shock proteins (Csps). The authors discussed the role of Csps (CspA, B, and D) in L. monocytogenes stress tolerance, i.e., enabling bacteria coping with a wide range of stress conditions such as cold, desiccation, osmotic, and oxidative stress. The authors also summarized the impact of Csps in L. monocytogenes motility, biofilm formation, and virulence. In addition, the authors discussed the hierarchical trends and functional redundancies of L. monocytogenes Csps and dependence of genetic background / strains on functions of Csps. This review addresses the role of Csps in L. monocytogenes physiology, ranging from stress responses to virulence regulation, which will provide helpful information to the readers in the field who are interested in this topic.

Overall, this review is comprehensive.  Here are some suggestions for authors to consider.

(1) The statements in line 57-59 are very speculative. The authors should either provide references or tone down the language.

(2) Line 167-169, the sentence need to be revised.

(3) Line 174: Do authors mean that all mutant strains were able to reach stationary phase?

(4) Line 180-183: Why do authors compare cspA transcription level to cspB and cspD?
